# An Unfavorable Outcome of *M. chimaera* Infection in Patient with Silicosis

**DOI:** 10.3390/diagnostics12081826

**Published:** 2022-07-29

**Authors:** Ewa Łyżwa, Izabela Siemion-Szcześniak, Małgorzata Sobiecka, Katarzyna Lewandowska, Katarzyna Zimna, Małgorzata Bartosiewicz, Lilia Jakubowska, Ewa Augustynowicz-Kopeć, Witold Tomkowski

**Affiliations:** 11st Department of Lung Diseases, National Research Institute of Tuberculosis and Lung Diseases, 01-138 Warsaw, Poland; i.siemion@igichp.edu.pl (I.S.-S.); m.sobiecka@igichp.edu.pl (M.S.); k.lewandowska@igichp.edu.pl (K.L.); k.zimna@igichp.edu.pl (K.Z.); m.bartosiewicz@igichp.edu.pl (M.B.); w.tomkowski@igichp.edu.pl (W.T.); 2Department of Radiology, National Research Institute of Tuberculosis and Lung Diseases, 01-138 Warsaw, Poland; l.jakubowska@igichp.edu.pl; 3Department of Microbiology, National Research Institute of Tuberculosis and Lung Diseases, 01-138 Warsaw, Poland; e.kopec@igichp.edu.pl

**Keywords:** *Mycobacterium chimaera*, nontuberculous mycobacterial disease, silicosis, extracorporeal membrane oxygenation

## Abstract

*Mycobacterium chimaera* is a slow-growing, nontuberculous mycobacterium (NTM) belonging to the *Mycobacterium avium complex* (*MAC*). It was identified as a unique species in 2004. Since 2013 it has been reported as a cause of disseminated infection in patients after cardiac surgeries. Only a few cases associated with underlying lung diseases have been noted. *M. chimaera* infection is characterized by ambiguous symptoms. There is no treatment with proven effectiveness, and it has a poor prognosis. Silicosis is a disease that can predispose to mycobacterial infection. Silica damages pulmonary macrophages, inhibiting their ability to kill mycobacteria. We present a case of *M. chimaera* infection in a patient with silicosis and without other comorbidities. To our knowledge, it is the first case of silicosis associated with *M. chimaera* disease. A 45-year-old man presented with a persistent low-grade fever. Based on the clinical and radiological picture, positive cultures, and histological examination, the nontuberculous mycobacterial disease was diagnosed. First, multidrug therapy according to the treatment guidelines for *MAC* was implemented, then antibiotics were administrated, based on drug sensitivity. Despite the treatment, eradication was not achieved and the patient died. The analysis of *M. chimaera* infection cases could contribute to developing recommendations and thus improve the prognosis.

## 1. Introduction

*Mycobacterium chimaera* is a slow-growing, nonpigmented, acid-fast positive, nontuberculous mycobacterium (NTM) belonging to the *Mycobacterium avium complex (MAC).* It is ubiquitous mycobacteria, often found in natural water sources. It was identified as a unique species in 2004. Since 2013 it has been reported as a cause of disseminated infection in patients after open-heart surgeries exposed to contaminated heater-cooler devices. The wide application of the extracorporeal membrane oxygenation (ECMO) system requires awareness of the possibility of *M. chimaera* infection. Interestingly, patients treated with ECMO because of cardiac surgeries are more likely to get infected with *M. chimaera* than those treated with ECMO due to respiratory failure because of larger potential entry sites for the pathogen [1]. 

Moreover, since 2013, some cases in immunocompromised individuals and patients with underlying lung diseases like tuberculosis, chronic obstructive pulmonary disease (COPD), or interstitial lung diseases have been described [2,3,4]. Furthermore, silicosis is a disease that can predispose to mycobacterial infection and is a pneumoconiosis caused by the inhalation of crystalline silicon dioxide. Tiny particles of silica are phagocyted by macrophages, leading to the accumulation of free radicals, which result in the release of inflammatory cytokines, increased cell signalling, and apoptosis of parenchymal cells and macrophages [5].

Moreover, surfactant protein A, which is elevated in bronchoalveolar lavage fluid of individuals with silicosis, restrains the activated macrophages’ creation of reactive nitrogen species and enables mycobacteria to enter the macrophages without inducing cytotoxicity [6]. Due to nonspecific symptoms and a long latency period, *M. chimaera* infections may not be diagnosed and treated promptly and thus can be life-threatening. The virulence and pathogenicity of *M. chimaera* are still debated. We present below a case report of *M. chimaera* infection in a patient with silicosis that, despite the treatment in accordance with international recommendations, had a negative outcome.

## 2. Case Report

A 45-year-old man, with a history of silicosis, after tonsillectomy in childhood and with no other comorbidities, presented in 2017 with persistent low-grade fever, increasing shortness of breath on exertion, night sweats, and weight loss. Silicosis was diagnosed three years earlier based on the histopathologic evaluation of an open lung biopsy. The patient presented then with similar symptoms accompanied by nonspecific chest pain that persisted after respiratory system infection. He worked on sandblasting metals in the past. Radiological examination (X-ray and computed tomography (CT)) (Figure 1 and Figure 2) showed diffused nodular opacities with consolidation in apical right lung segments and lymphadenopathy. 

Tuberculosis, sarcoidosis, and silicosis were considered in the differential diagnosis. Sputum acid-fast bacilli (AFB) smears and cultures were negative. Bronchoscopy was performed, AFB smears and cultures were negative, and the bronchoalveolar lavage fluid microscopic evaluation results were inconclusive. Open lung biopsy was then performed, and silicosis was diagnosed. Radiological examinations revealed slow progression of parenchymal consolidations, most intensified in the right lung. Severe restriction with moderately decreased transfer factor for carbon monoxide (TL_co_) was observed in pulmonary function tests (PFTs) (Table 1), and the 6-min walking test results were normal.

In 2017, the patient was hospitalized in a hematological department due to axillary lymphadenopathy. The axillary lymph node biopsy was performed. The neoplasm was ruled out. Afterwards, the patient was admitted to our department due to worsening of all constantly observed symptoms mentioned above. Laboratory tests showed moderately elevated c-reactive protein (CRP)—13 mg/L (N: <5 mg/L) and D-dimers level—641 ng/mL (N: <500 ng/mL); blood cell count, liver enzymes, electrolytes, creatinine, coagulation parameters, and blood gases were within normal limits. Posteroanterior X-ray (Figure 3) showed progression of disseminated lung lesions, large opacities, and conglomerate masses in the upper and middle zones with retraction of hila. Computed tomography pulmonary angiography (CTPA) (Figure 4) scans excluded pulmonary embolism and showed progression of the previously described bilateral parenchymal changes in the apical lung segments and lower lung lobes lymphadenopathy and disseminated nodular changes. 

The lymph node biopsy specimen was revised in our hospital. The repeated histological evaluation revealed epithelial necrotizing granulomas. AFB smears and specimen cultures were negative. Twelve samples of the sputum were examined for tuberculosis and mycobacteria. Four of them obtained the growth on Middlebrook liquid medium in the Bactec MGIT system. The Ziehl-Neelsen staining of smear from culture revealed acid-fast mycobacteria. The TBC ID MGIT identification test based on protein MPT64 production was performed, and the organism was preliminarily identified as NTM. Species identification by genetic test (GenoType Mycobacterium CM VER 2.0 and GenoType Mycobacterium NTM-DR) verified the obtained culture as *Mycobacterium chimaera* (Figure 5 and Figure 6). 

Mycobacterial disease was diagnosed based on the clinical and radiological picture, positive cultures, and histological examination. Antimicrobial, multidrug therapy was administrated according to the treatment guidelines for the *Mycobacterium avium complex*- Clarithromycin 500 mg every 12 h, Rifampicin 600 mg per day, and Ethambutol 1000 mg per day. An ophthalmological examination was performed before Ethambutol application. Cultures taken after 3 and 9 months were negative. The patient’s condition improved, results of radiological and functional tests were stable. The total treatment length was 15 months. After the therapy was finished patient’s condition gradually deteriorated, and slow radiological progression was observed. Sputum cultures grew *M. chimaera* again. Antibiotics were then administrated based on drug sensitivity tests from the initial sputum cultures. *M. chimaera* was resistant to most medications. The treatment included Rifabutin 300 mg daily and Clofazimine 100 mg twice a day. Despite the treatment, eradication was not achieved. The patient’s condition gradually deteriorated and after another year of treatment, the patient died.

## 3. Discussion

Nowadays, an increased number of isolates of NTM has been noticed. *Mycobacterium kansasi*, *Mycobacterium avium*, and *Mycobacterium xenopi* belong to the most often recognized and well-known mycobacteria [7]. Initially, the enormous interest in *Mycobacterium chimaera* was associated with patients who underwent cardiac surgeries with extracorporeal circulation [8,9,10,11]. After three months to five years after surgery, patients presented symptoms similar to those seen in the disseminated mycobacterial disease, like persistent dry cough, low-grade fever, asthenia, night sweats, and weight loss. Some of them reported fever, chest or abdominal pain, somnolence, and dysarthria. Moreover, one case of *M. chimaera* infection has been described in a man that has never undergone an open-heart procedure but only worked in the past in operation theatre, where they took place [12]. What is noticeable is that it is very difficult to find an association with the undergone procedure because of the long time to clinical manifestation. The incubation period after exposure to *M. chimaera* is usually 3–72 months [1]. Disease caused by *M. chimaera* can be limited to the lungs, but in cases related to open-heart procedures, it is often disseminated. Evidence suggests that *M. chimaera* may be a causative agent of valve prosthesis endocarditis, ocular congestion, osteomyelitis, hepatitis, and renal dysfunction, as well as other life-threatening conditions [13,14,15]. It is important to pay attention to the patient’s past medical history. In the presented case, the patient had no history of cardiac surgery, which was precisely verified on admission. Echocardiography and abdominal ultrasound were performed several times during observation in our department and no abnormalities were found. Further examinations were not required, considering the lack of clinical symptoms of disseminated disease. 

*M. chimaera* infections are not always related to a history of cardiac surgery; however, they are almost every time associated with some comorbidities. COPD, previous tuberculosis, and interstitial lung diseases seem to be the most often noted in these patients [4]. The literature provides an insufficient number of case reports where the clinical outcome was precisely evaluated, which makes our publication noteworthy. In one of them, symptoms reported by a patient with COPD were typical of mycobacterial infection, and after clinical assessment, the authors did not administrate any therapy [2]. In another article, an immunocompromised individual with lymphoma presented with the disseminated disease and received multidrug therapy. Lymphoma treatment was modified due to *M. chimaera* symptoms, disease progression was observed, and this patient died [3]. 

Silicosis is one of the illnesses predisposing to mycobacterial disease. It is thought that silica damages pulmonary macrophages, inhibiting their ability to kill mycobacteria. To our knowledge, it is the first case of silicosis associated with *M. chimaera* disease, which makes our article unique and noteworthy. Infections due to *M. chimaera* are considered rare; however, their prevalence may be underestimated. Zabost et al. reported over 86 patients from one department of microbiology, whose diagnoses were changed after repeated examination, including modern gene analysis [4]. According to ATS/IDSA recommendations for recognition of nontuberculous mycobacterial disease (Table 2)—clinical and at least one of the microbiological criteria, including some kind of positive culture, are necessary to recognize the disease [16].

Lung mycobacterial disease diagnostics require imaging examinations, where consolidations, excavations, and necrotizing pneumonia are typical findings [1]. *M. chimaera* can be grown using standard culture methods. The growth time and colony morphology are identical to *Mycobacterium intracellulare—*it grows slowly (6–8 weeks) at temperatures 25–35 °C, and the colonies are smooth and not pigmented [1,4]. *M. chimaera* is closely related to *M. intracellulare*. According to Validation list no. 148, the name is *M. intracellulare* subsp. *chimaera* [17]. They show only one nucleotide difference in the 16S ribosomal DNA sequences. *M. chimaera* can be misidentified as *M. intracellulare* by mass spectrometry (MALDI-TOF MS) or some commercial DNA hybridization probe assays [18]. The preferred method that allows distinguishing these two species is nucleic acid sequencing with 16-23S rRNA region analysis, which was unavailable before 2004 [1,4]. The Accu Probe and Lipav 1 tests were then commonly used, and they did not allow for the identification of *M. chimaera* within *MAC* species; there are also some commercial methods accessible as an alternative [19]. It is noteworthy that atypical mycobacterial infections can cause both caseating and noncaseating granulomas. Granulomas in disseminated disease were found in cardiac tissue, liver, hemispheres and brain stem, kidneys, and bone marrow [10]. In some cases, an incorrect diagnosis of sarcoidosis was made after considering exclusively histological examination results [1,9]. It could cause essential morbidity in sarcoidosis because steroids used as first-line treatment can worsen *M. chimaera* infection. The treatment of *M. chimaera* disease is not clearly established. As in other mycobacterial infections, the treatment is not mandatory and depends on the disease course. In case of clinical or radiological deterioration, the patient may require prolonged multidrug therapy to control the infection. The majority of patients get antibiotic therapy according to the treatment guidelines for the *Mycobacterium avium complex—*macrolide (azithromycin 250–500 per day or clarithromycin 500 mg every 12 h), rifampicin 10 mg/kg/day, and ethambutol 15–25 mg/kg/day [1,20]. Macrolide susceptibility testing is required, and patients should be cautiously monitored because of the possibility of developing macrolide resistance. The treatment length is at least 12 months after sputum conversion in case of lung disease. There are limited data from clinical experience in *M. chimaera*. The multidrug therapy mentioned above was also administered to our patient and was ineffective. Wild type *M. chimaera* is usually susceptible to clarithromycin; however, resistant isolates were also reported, especially after previous antibiotic therapy. Mok et al. mentioned that 18% of their probe was resistant to rifampicin and 11% to ethambutol, however, susceptibility to these antibiotics is not routinely checked in *MAC*, because in vitro results are not always reliable compared to clinical response [21]. In our patient rifabutin, clofazimine, ethambutol and amikacin were administrated when culture grew *M. chimaera* after first-line therapy. In Mok’s probe, only 2% of mycobacteria were not fully susceptible to rifabutin and amikacin. Moreover, clofazimine and amikacin showed significant synergistic activity against *MAC* strains in vitro, making them important in *M. chimere* treatment [21]. The outcome of the treatment in our patient was negative. Poor prognosis has been mentioned in the literature. Despite low pathogenicity, mortality in *M. chimaera* remains high at 50–60% [1,9].

## 4. Conclusions

In patients with silicosis and symptoms that suggest infection, mycobacterial disease should be considered in the differential diagnosis. *M. chimaera* infection is characterized by ambiguous symptoms. Moreover, the course of the disease, as in our case, can be prolonged. Diagnosis requires using modern genetic techniques and not all tests available in various laboratories are specific enough. Moreover, there is no treatment with proven effectiveness and even proceeding according to the guidelines, as it was in our patient, eradication is not always achieved. The disease has a poor prognosis despite the treatment. Detailed analysis of different patients’ management could lead to the development of diagnostic and therapeutic recommendations for *M. chimaera* infection.

## Figures and Tables

**Figure 1 diagnostics-12-01826-f001:**
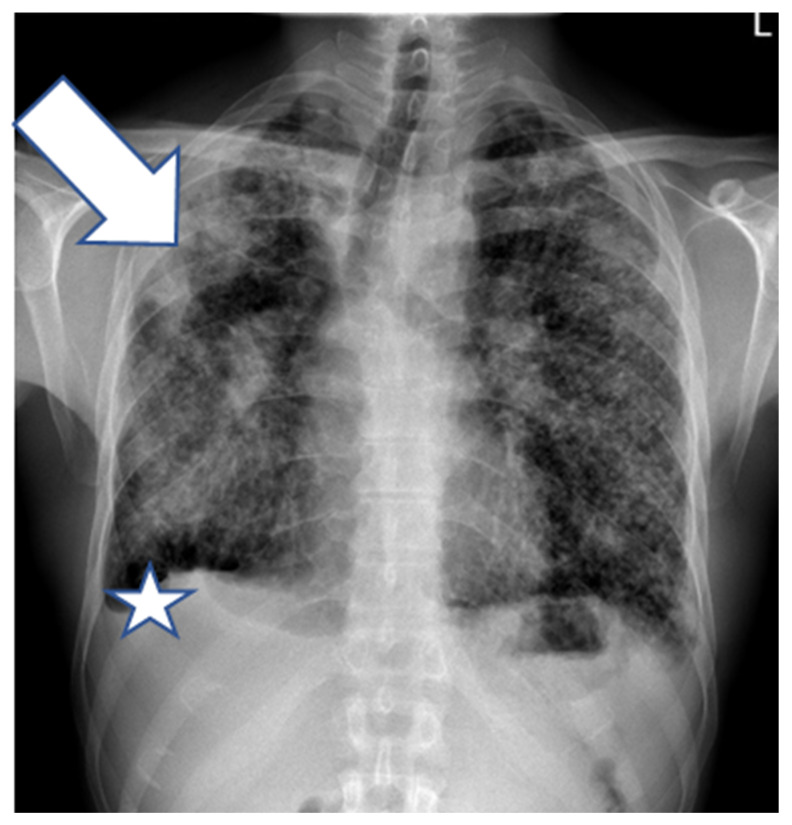
Posteroanterior chest X-ray (2015) shows multiple small diffuse well-defined nodules, confluent opacities in the upper zones and the middle right zone (arrow), hilar lymphadenopathy, and small right-sided pleural effusion (asterisk).

**Figure 2 diagnostics-12-01826-f002:**
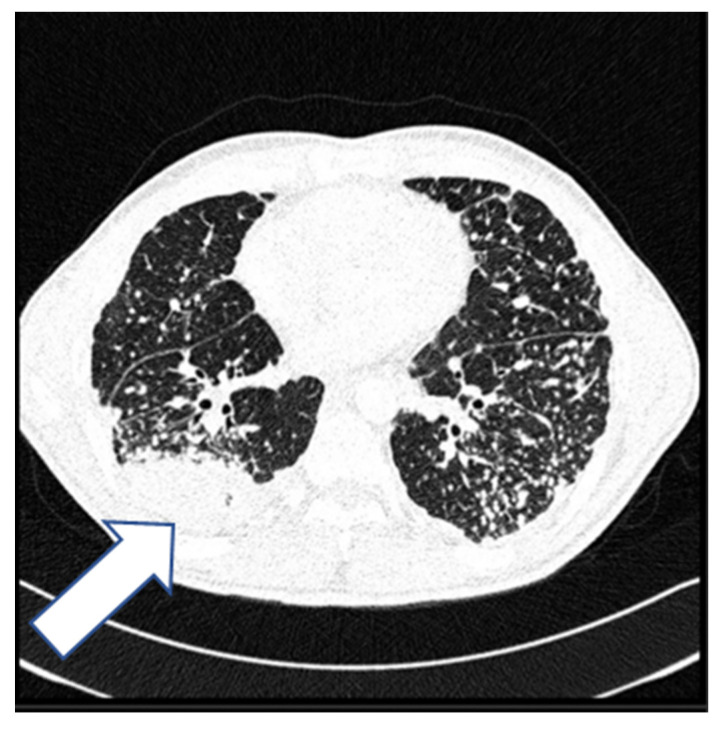
High-resolution computed tomography of the lungs (2015) shows numerous, small, well-defined nodules with a perilymphatic distribution and consolidations in the lung periphery (arrow).

**Figure 3 diagnostics-12-01826-f003:**
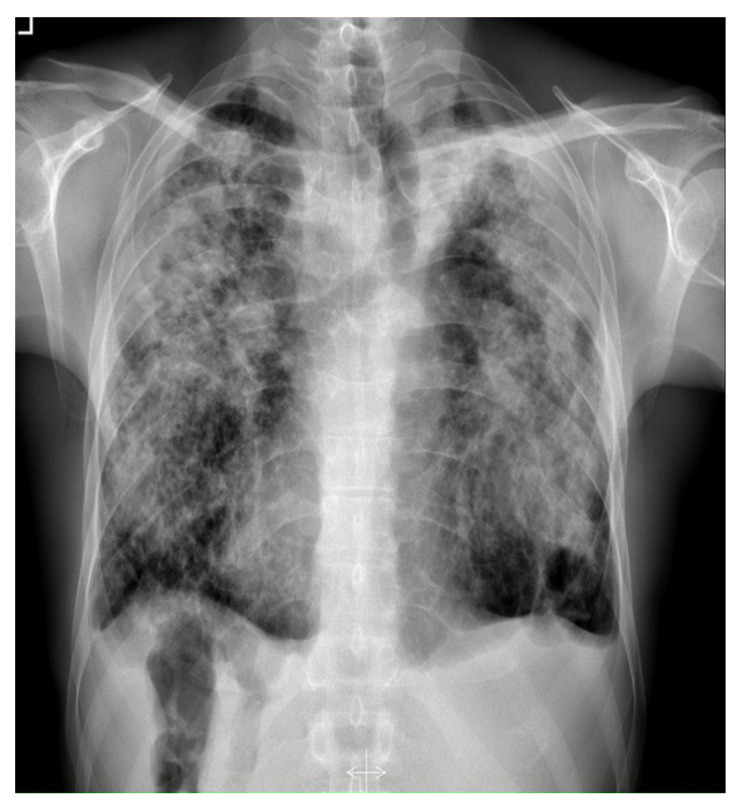
Posteroanterior chest X-ray (2017) shows the evident progression of disseminated lung lesions, large opacities, and conglomerate masses in the upper and middle zones with retraction of hila.

**Figure 4 diagnostics-12-01826-f004:**
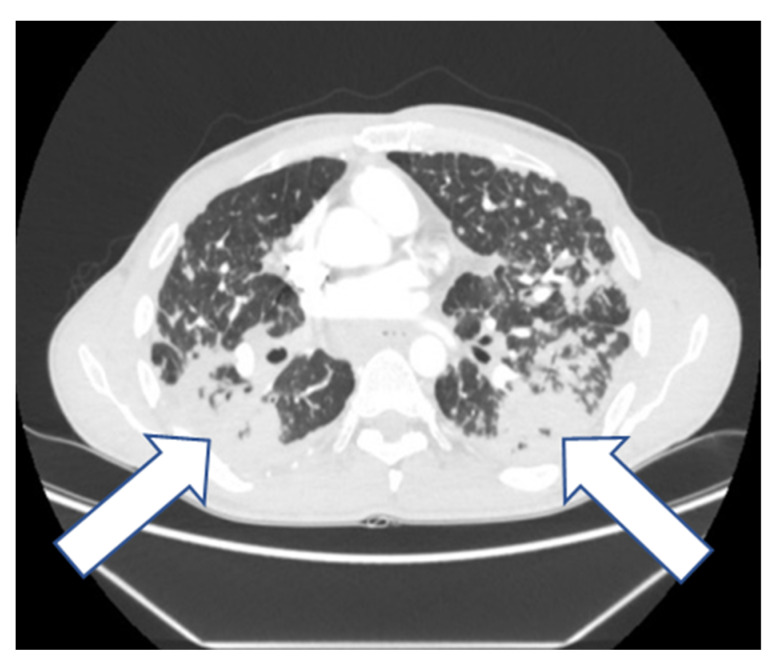
CT scan (2017) shows diffuse nodules and bilateral conglomerate masses (arrows) associated with distortion of lung architecture.

**Figure 5 diagnostics-12-01826-f005:**
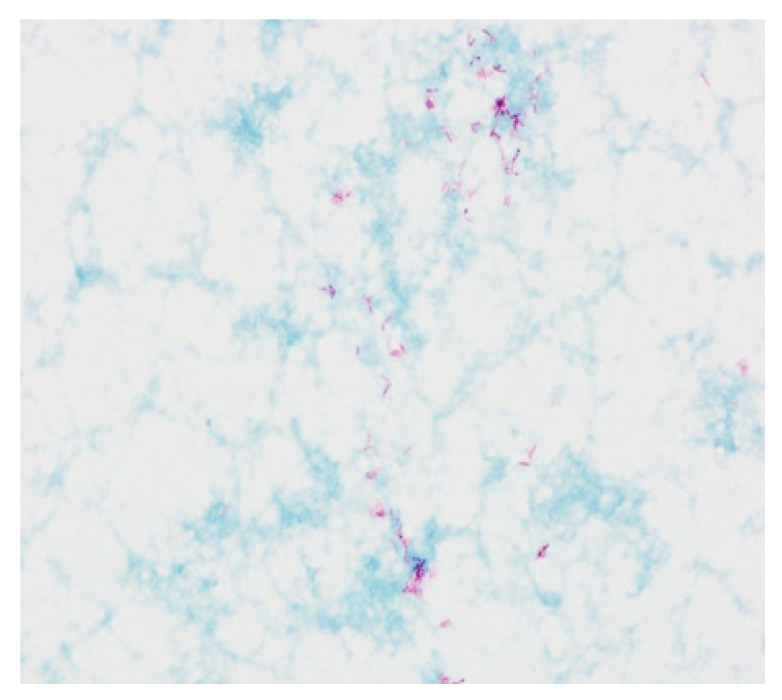
The acid-fast rods of mycobacterium. Smear made from a colony, Ziehl-Neelsen stain.

**Figure 6 diagnostics-12-01826-f006:**
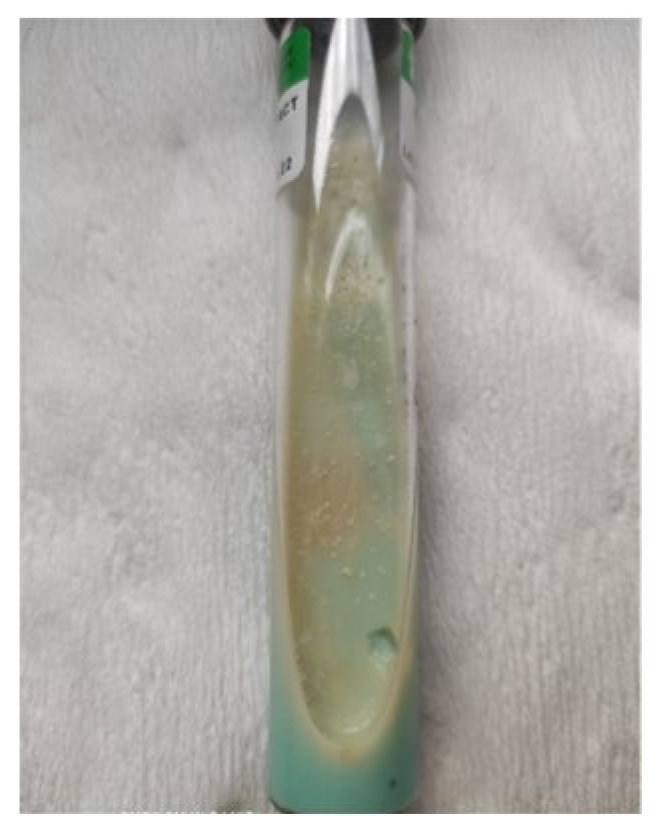
The colony growth Mycobacterium chimaera on Löewenstein-Jensen solid medium.

**Table 1 diagnostics-12-01826-t001:** Pulmonary function tests (PFTs).

	Pred		Act1	%Pred	SR
Date Time			15-02-02 08:57:42		
FEV 1% VC MAX FEV 1% FVC	[%] [%]	79.83 81.76	84.26 84.26	105.5 103.1	0.62 0.42
VC MAX FVC	[L] [L]	4.63 7.79	2.14 2.14	46.2 44.7	−4.47 −4.28
FEV 1 MMEF 75/25 PEF	[L] [L/s] [L/s]	3.87 4.25 8.89	1.80 1.72 5.76	46.6 40.4 64.9	−4.70 −2.44 −2.59
FET V backextrapolation ex	[s] [L]		7.16 0.12		
R tot ITGV RV TLC RV % TLC	[KPa*s/L] [L] [L] [L] [%]	0.30 3.28 1.91 6.58 29.95	0.28 2.47 1.70 3.84 44.27	93.7 75.3 88.9 58.3 147.8	−1.35 −0.52 −3.93 2.63
DLCOc SB DLCOcNA VIN VA	[mmol/min/kPa] [mmol/min/kPa/L] [L] [L]	10.26 1.56 4.63 6.43	5.57 1.81 2.02 3.08	54.3 116.0 43.6 47.9	−3.34 −4.68

**Table 2 diagnostics-12-01826-t002:** ATS/IDSA recommendations for recognition of nontuberculous mycobacterial disease based on ref. [16].

Clinical symptoms (any of the following)	pulmonary—including but not limited to: cough, sputum, hemoptysis
	systemic—including but not limited to: fever, weight loss, sweats
Radiologic presentation (any of the following)	X-ray: nodular or cavitary lesions
	HRCT: bronchiectasis and nodular opacities
Microbiologic tests (any of the following)	positive culture results from at least two separate sputum samples
	positive culture result from at least one bronchial wash or lavage
	histopathological features of mycobacterial disease (granulomas or AFB) and positive culture for NTM OR histopathological features of mycobacterial disease and one or more cultures positive for NTM from sputum or bronchial washings
AND:	Exclusion of other diseases

## Data Availability

Not applicable.

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
