# Peer review of "An Unfavorable Outcome of M. chimaera Infection in Patient with Silicosis"

_diagnostics, 2022, doi:10.3390/diagnostics12081826_

Round 1

Reviewer 1 Report

This is an interesting manuscript on the relevance and difficulties that NTM mycobacterial infections represent for clinicians. The manuscript is sound and deserves publication.

The main message is clear: Mycobacterium chimaera infections should be considered in patients with long-lasting clinical symptoms, mainly when comorbidities are present.

This reviewer has a single comment on the antimicrobial therapy administered just after mycobacterial disease was diagnosed (lines 108 and following). Was the strain checked for resistance/susceptibility test? Or just treatment guidelines were applied. This point is not indicated and could be relevant when taking into consideration the final unfavorable result.

Authors should be aware that changes in the taxonomic status has been described for M. chimaera. This species, after described in 2004 (Tortoli and co-workers) was identified as a subspecies of M. intracellulare  in 2018 (see Nouioui et al, Frontiers Microbiology 9:2007; and validated as “M. intracellulare  subspecies chimaera in the list number 148, see IJSEM 68:3379-3393). According to the previous statement, authors should modify the lines 154 and followers and mention that M. chimaera is so close to M. intracellulare  because they actually are the same bacterial species. Nonetheless, the name M. chimaera can be used as a taxonomic homotypic synonym of the validated name (https://lpsn.dsmz.de/species/mycobacterium-chimaera)

Author Response

We thank the reviewer for the article evaluation and all the comments. We took all of them into consideration and we believe our manuscript is improved as a result. 

This reviewer has a single comment on the antimicrobial therapy administered just after mycobacterial disease was diagnosed (lines 108 and following). Was the strain checked for resistance/susceptibility test? Or just treatment guidelines were applied. This point is not indicated and could be relevant when taking into consideration the final unfavorable result.

  • When the mycobacterial disease was diagnosed antimicrobial, multidrug therapy was administrated just according to the treatment guidelines for the Mycobacterium avium complex. Susceptibility tests were nonetheless performed. When sputum cultures grew M. chimaera again, antibiotics were administrated based on drug sensitivity from these initial cultures. M. chimaera was there resistant to most medications. The information has been now placed on lines 126-128.

Authors should be aware that changes in the taxonomic status has been described for M. chimaera. This species, after described in 2004 (Tortoli and co-workers) was identified as a subspecies of M. intracellulare  in 2018 (see Nouioui et al, Frontiers Microbiology 9:2007; and validated as “M. intracellulare  subspecies chimaera” in the list number 148, see IJSEM 68:3379-3393). According to the previous statement, authors should modify the lines 154 and followers and mention that M. chimaera is so close to M. intracellulare  because they actually are the same bacterial species. Nonetheless, the name M. chimaera can be used as a taxonomic homotypic synonym of the validated name (https://lpsn.dsmz.de/species/mycobacterium-chimaera)

  • We took into account the information about the taxonomic status. It can be found on line 180. In line with your suggestion, the name  chimaera can be used as a synonym of the validated name, and moreover, is widely used in literature so we decided to leave this form in the article.

Reviewer 2 Report

This topic is of interest and appropriate for the Journal but both presentation and discussion of data need to be revised to better perceive the scientific sound of this study.

Accordingly, I recommend this manuscript entitled “An unfavorable outcome of M. chimaera infection in patient with silicosis” for publication only after it is minor revised according to the indicated amendments. In the following, I present some issues that justify my decision.

Abstract:

Abstract reflects the whole manuscript. Abstract is not quite concise, rewrite the abstract, better start with study background (briefly) then discuss your methodology and results with discussion. 

Introduction: 

Introduction not well presented. Please add epidemiological data, silicosis definition etc. 

ü  I suggest the authors to write more from recent research articles about silicosis and pathophysiology of Mycobacterium chimaera.

ü  Some earlier reports are present in context to the proposed work by the authors. The authors must discuss those articlesThe authors are advised to put a closer look at these studies and explain how their work is different.

ü  Virulence and pathogenicity of M. chimaera are still debatedthis information may be succinctly added to the introduction, when describing the facts.

Line 50: A 45-year-old man, with a history of silicosis, and worked on sandblasting metals in the past. I’m concerned about his cardiac disorder, did authors check patient history, any evidence of prosthetic valve/graft infection/cardiovascular surgery/cardiopulmonary bypass, checked through “Echocardiography/positron emission tomography/computed tomography (PET/CT). Please provide more data about comorbidity. 

Line 100: Please provide more details of Species identification by genetic test.

·       Any data about pulmonary exacerbation_Forced expiratory volume (FEV1) or forced vital capacity (FVC) 

Line 142: It is thought that silica damages pulmonary macrophages, inhibiting their ability to kill mycobacteria_Please add more about pathophysiology 

Line 191: Conclusion is very short. Some details about the outcome of the work should be included.

Author Response

We thank the reviewer for the article evaluation and all the critical remarks. We took all of them into consideration, we reorganized all parts of the article and added necessary information to the text. We believe our manuscript is improved as a result.

Abstract:

Abstract reflects the whole manuscript. Abstract is not quite concise, rewrite the abstract, better start with study background (briefly) then discuss your methodology and results with discussion. 

  • The whole abstract has been changed. You can find it at the beginning of the article. There is now more general information and fewer patient medical history details. 

Introduction

Introduction not well presented. Please add epidemiological data, silicosis definition etc. 

  1. I suggest the authors to write more from recent research articles about silicosis and pathophysiology of Mycobacteriumchimaera.
  • We added some general information about silicosis and mentioned the correlation with mycobacterial lung disease. Pathophysiology has also been discussed. We considered 2 more literature positions. You can find it on lines 44-53.
  1. Some earlier reports are present in context to the proposed work by the authors. The authors must discuss those articles. The authors are advised to put a closer look at these studies and explain how their work is different.
  • We have discussed available articles. The literature is insufficient in case reports about chimaera infection outcomes and a few details are described. This fact shows that our article is unique and noteworthy. You can find this information on lines 157-163
  1. Virulence and pathogenicity of  chimaeraare still debated, this information may be succinctly added to the introduction, when describing the facts.
  • The sentence has been added, you can find it on line 55.
  1. Line 50: A 45-year-old man, with a history of silicosis, and worked on sandblasting metals in the past. I’m concerned about his cardiac disorder, did authors check patient history, any evidence of prosthetic valve/graft infection/cardiovascular surgery/cardiopulmonary bypass, checked through “Echocardiography/positron emission tomography/computed tomography (PET/CT). Please provide more data about comorbidity. 
  • We added the information that our patient definitely had no history of cardiac disorders. We also described other examinations which were performed on our patient. You can find it on lines 150-153. Moreover, we added information that apart from tonsillectomy in childhood, the patient had no other comorbidities. You can find it now at the beginning of case report line 59.

  1. Line 100: Please provide more details of Species identification by genetic test.
  • We added information about one more test that was performed on our patient- line 111. Moreover, we discussed tests used in the past for mycobacterial species identification- line 186.

  1. Any data about pulmonary exacerbation_Forced expiratory volume (FEV1) or forced vital capacity (FVC) 
  • We added the whole pulmonary function test figure (Fig. 3.)

  1. Line 142: It is thought that silica damages pulmonary macrophages, inhibiting their ability to kill mycobacteria_Please add more about pathophysiology 
  • We discussed it in an introduction as I mentioned above. There were 2 additional articles taken into consideration.

  1. Line 191: Conclusion is very short. Some details about the outcome of the work should be included.
  • We added some context to the conclusion and we considered our patient’s case in accordance with the described in the whole article informations.
